# Biological Factors behind Melanoma Response to Immune Checkpoint Inhibitors

**DOI:** 10.3390/ijms21114071

**Published:** 2020-06-06

**Authors:** Magdalena Olbryt, Marcin Rajczykowski, Wiesława Widłak

**Affiliations:** Maria Skłodowska-Curie National Research Institute of Oncology, Gliwice Branch, Wybrzeze Armii Krajowej 15, 44-102 Gliwice, Poland; Marcin.Rajczykowski@io.gliwice.pl (M.R.); Wieslawa.Widlak@io.gliwice.pl (W.W.)

**Keywords:** melanoma, immunotherapy, immune checkpoint inhibitors, biomarkers of response

## Abstract

Modern immunotherapy together with targeted therapy has revolutionized the treatment of advanced melanoma. Inhibition of immune checkpoints significantly improved the median overall survival and gave hope to many melanoma patients. However, this treatment has three serious drawbacks: high cost, serious side effects, and an effectiveness limited only to approximately 50% of patients. Some patients do not derive any or short-term benefit from this treatment due to primary or secondary resistance. The response to immunotherapy depends on many factors that fall into three main categories: those associated with melanoma cells, those linked to a tumor and its microenvironment, and those classified as individual ontogenic and physiological features of the patient. The first category comprises expression of PD-L1 and HLA proteins on melanoma cells as well as genetic/genomic metrics such as mutational load, (de)activation of specific signaling pathways and epigenetic factors. The second category is the inflammatory status of the tumor: “hot” versus “cold” (i.e., high versus low infiltration of immune cells). The third category comprises metabolome and single nucleotide polymorphisms of specific genes. Here we present up-to-date data on those biological factors influencing melanoma response to immunotherapy with a special focus on signaling pathways regulating the complex process of anti-tumor immune response. We also discuss their potential predictive capacity.

## 1. Immunotherapy of Melanoma 

Immunotherapy based on immune checkpoints inhibition (as well as targeted therapy) is nowadays a standard treatment for patients with advanced melanoma. Modern immunotherapy takes advantage of the inhibition of particular immune checkpoint molecules present on immune cells (cytotoxic T-lymphocyte antigen 4, CTLA-4, programmed cell death 1 protein, PD-1) or tumor cells (PD-L1) with specific monoclonal antibodies and currently is used for treatment not only of advanced melanoma, but also lung cancer, renal cancer, and others [1]. Immune checkpoint inhibition results in the reactivation of immune response against tumor cells by releasing T cells from the inhibitory effect of these checkpoint molecules. The scientific discoveries that paved the way to this modern immunotherapy were appreciated by awarding the Nobel Prize in Medicine or Physiology 2018 jointly to James P. Allison (MD Anderson Cancer Center, Houston) and Tasuku Honjo (Kyoto University) for the discovery of CTLA-4 and PD-1, respectively. Currently, there are three agents approved for advanced melanoma treatment: ipilimumab (anti-CTLA-4; since 2011), nivolumab, and pembrolizumab (both anti-PD-1; since 2014). These drugs are also registered as a therapy in III-stage melanoma [2] in adjuvant scheme. Other agents and combinations are currently in clinical trials, e.g., PD-L1, LAG3, IDO, and HDAC inhibitors as well as oncolytic viruses [3]. Modern therapies significantly improved the survival of advanced melanoma patients [4]. However, not all patients derive benefit from immune checkpoint inhibitors (ICIs) treatment. Approximately 50% of patients do not respond to monotherapy, 40% are not responsive to combined immunotherapy and approximately 30% of the responders experience disease relapse after initial response. This is because of primary (intrinsic) and secondary (acquired) resistance to ICIs. Immune checkpoint inhibition has also other limitations. Patients are exposed to immune-related adverse events such as colitis, hypothyroidism, hepatitis, hypophysitis, hyperthyroidism, and pneumonitis. They are significantly escalated in combined anti-CTLA-4 and anti-PD-1 therapy [5]. Immunotherapy is also one of the most expensive treatments which not all public health systems can afford. Therefore, it is so crucial to elucidate the mechanisms of primary and acquired resistance to immunotherapy and develop a predictive test that would enable us to estimate the efficacy of this treatment in individual patients before its initiation, and qualify for treatment only those patients with the highest probability of response. The anti-tumor immune response is determined by many factors that can be divided into three groups depending on the anatomical level: those associated with melanoma cells, those linked to the tumor microenvironment, and those determined by the whole organism (Figure 1). Most of them are associated with or are dependent strictly on melanoma cells and their aberrant molecular signaling. 

## 2. PD-1/PD-L1 Signaling as an Immunotherapy Target 

Although both PD-1 and CTLA-4 function as immune checkpoints and downregulate immune responses, anti-PD-1 therapy has been proved to be more effective in the extension of progression free survival (PFS) (nivolumab, pembrolizumab) and overall survival (OS) (pembrolizumab) than anti-CTLA-4 treatment with ipilimumab [6,7]. Thus, anti-PD-1 agents are the first-choice immunotherapy for advanced melanoma treatment, which can be optionally combined with ipilimumab (as combined treatment prolongs PFS) [7]. Programmed cell death 1 receptor (PD-1, also known as PDCD1, CD279) is a checkpoint molecule present on T cells, B cells, and NK cells, which can interact with its ligands: PD-L1 (CD274; expressed on tumor cells, fibroblasts, antigen presenting cells, and others) or PD-L2 (PDCD1LG2; present mainly on hematopoietic cells). Ligation of PD-1 with its ligand leads to T cell dysfunction, neutralization and anergy, which is why overexpression of PD-L1 by tumor cells attenuates anti-tumor response [8]. Intuitively, patients with an elevated level of this protein in the tumor should have the best response to this therapy. Indeed, patients with overexpression of this ligand on melanoma cells are more likely to benefit from therapy with nivolumab or pembrolizumab than those with low or no expression of the ligand [9,10,11]. Also, amplification of *PD-1* and *PD-L1* in some cases leads to a durable response to immunotherapy [12]. However, despite numerous studies in melanoma, lung cancer, and renal cancer, no universal, predictive test based on PD-L1 expression has been developed so far. In 2015, FDA approved an immunohistochemical test for PD-L1 evaluation (28-8 pharmDx) in lung cancer treatment with nivolumab, and subsequently in 2016, a similar test (22C3 pharmDx) in melanoma treatment (it is also used in some clinical trials for patient recruitment; e.g., in NCT03829332 study). However, these tests, to our knowledge, have not entered clinical practice. One of the main hurdles for their usage is the establishing of clinically valid cut-off points based on the percentage of tumor cells with PD-L1 expression in the tumor [13]. 

Most studies show that irrespective of the defined cut-off points (e.g., 1% or 5% of tumor cells expressing PD-L1) a substantial percentage of patients would be improperly qualified to the therapy. Up to 20% of ”non-responders“ respond to the treatment, while up to 50% of ”responders“ do not derive any clinical benefit from this therapy but suffer from side effects [14]. The KEYNOTE 001 clinical trial study showed that patients with PD-L1 expression in more than 10% of melanoma cells are more likely to respond to pembrolizumab treatment. However, approximately 10–20% of patients with lower expression also benefited from this treatment. Other studies confirm these observations. Espinoza et al. showed that PD-L1-positive patients had 50% chance of response, while in the PD-L1-negative group approx. 15% of patients also responded to anti-PD-L1 treatment [15]. 

The aforementioned results suggest that some melanomas are inherently resistant to immunotherapy irrespective of the PD-L1 status (primary resistance), while others respond to immune checkpoint inhibitors despite low PD-L1. It is not surprising considering the complex process of the anti-tumor immune response, which depends on many factors associated not only with tumor cells but also tumor microenvironment and the whole organism. All these interconnected factors influence the three main prerequisites for efficient anti-tumor immune activity, which are infiltration of the tumor with active and functional immune cells, recognition of tumor cells by immune cells [16], and apoptosis of tumor cells induced by immune cells [17]. Recognition of tumor cells by immune cells depends on the presence of tumor antigens and the process of antigen presentation to dendritic cells in the context of HLA proteins [16]. Infiltration of the tumor with immune cells and apoptosis are regulated by genetic and genomic determinants of cancer cells as well as tumor microenvironment and organism-associated factors, e.g., microbiome [8]. 

## 3. Tumor Mutational Burden (TMB) as an Indicator for Predicting Response to Immunotherapy

Elimination of tumor cells by the immune system takes place upon recognition of their “alien” peptides in the context of HLA proteins. This process is strictly dependent on the presence of tumor-specific antigens (TSA), which appear on tumor cells due to the mutational process [18]. A lack or a low number of these neoantigens can be caused by a low number of mutations in tumor cells, while a high number of mutations (>10/Mb) increases the chance of the appearance of new epitopes recognizable to the immune system [19]. Melanoma and lung cancer are the most mutated cancers [20], which is why patients suffering from these diseases benefit from immunotherapy to a higher extent than other cancer patients [21]. In melanoma, some UV-induced DNA damage is also prognostic for outcome [22]. 

The number of nonsynonymous, somatic mutations identified per megabase of the genome coding area in tumor cells (i.e., tumor mutational burden/load, TMB) correlates with the response to immunotherapy and some studies suggest that TMB may be an indicator for patients’ response to immunotherapy [23]. The analysis performed on 1662 patients with various cancers treated with immunotherapy revealed that for all cancers (except glioma) the TMB status correlated with therapy response and overall survival. Patients with the highest number of mutations had the best response rate and lived longer [24]. The TMB evaluation was based on analysis of ~3% of coding sequences using MSK-IMPACT assay (Integrated Mutation Profiling of Actionable Cancer Targets) and the cut-off points were adjusted individually to each tumor type. The patients classified to the “highly mutated” group were more likely to respond to immunotherapy, although approx. 40% of them did not derive any benefit from the treatment. The results suggest that TMB has potential predictive value, however, the cut-off points should be optimized to each cancer due to inter-cancer variability in TMB [21]. Most studies confirm this notion also for melanoma. An analysis of 151 samples of immunotherapy-treated patients (mainly melanoma and lung cancer) showed that higher TMB (>20 mutations/Mb) was independently associated with better response rate [25]. Most patients in the high TMB population (58%) derived benefit from anti-PD-1/PD-L1 therapy in comparison to 20% of patients in the low TMB group. Similarly, a whole exome sequencing (WES) analysis performed on a cohort of 110 melanoma patients revealed that both mutational load and neoantigens load correlate with response to ipilimumab [26]. The correlations were statistically significant, even though there were some low mutation load outliers in the responders’ group and high mutation load outliers in non-responders. The same was observed for the neoantigens load. Similar results have been previously obtained by Snyder et al. [27]. Genetic analysis of 64 samples (discovery and validation sets) showed that patients with long-term benefit from anti-CTLA-4 treatment had a significantly higher amount of exonic mutations than low-responders (and those with more than 100 mutations gained survival benefit). This is similar in comparison to Van Allen’s results in which both groups contained outliers. The difference in the number of mutations between non-responders and responders (anti-PD-1 therapy) were also observed by Hugo et al., while in this study the statistical significance was gained only for overall survival but not for response to immunotherapy [28]. The predictive capacity of TMB was also undermined by Wood at al. [29]. The analysis of 457 publicly available samples revealed no predictive value of TMB in renal cancer and very weak in melanoma and lung cancer treated with anti-CTLA-4 drugs. Incorporation of metrics associated with antigen presentation (proteasomal cleavage, transport, and surface presentation) did not improve the predictive capacity of TMB. All the aforementioned studies suggest that TMB has limited predictive capacity. Clinical utilization of this parameter, as a single metric may result in unintended harm due to the omission of therapy for patients with “low” TMB who might nonetheless benefit from it or due to the risk of toxicity in patients who would qualify to this therapy but despite “high” TMB would not benefit. A large number of mutations does not guarantee therapeutic success as the most crucial is not the number itself but the presence of novel antigens recognized by immune cells [27]. In all mentioned above studies the neoepitopes burden tended to correlate with response to immunotherapy, but only Snyder et al. [27] claimed to identify neoepitopes signature associated with clinical benefit from anti-CTLA-4 therapy. It seems that the vast majority of neoantigens arising from nonsynonymous mutations are patient-specific events. This is supported by in silico analyses. Efremova et al. revealed that only approximately 4% of new epitopes are common to two or more patients and only 7.6% come from mutations in the main cancer-related genes (mainly *BRAF*, *RAS*, and *PIK3CA*) [30]. 

Summing up, TMB may have predictive capacity in melanoma immunotherapy, but obviously this is not the only factor affecting anti-tumor response. Equally important is the capacity for presentation of neoantigens by the MHC system. The number of mutations and neoantigens potentially recognized by immune cells may be irrelevant if the antigen-presenting system does not work properly. 

## 4. Antigen-Presenting System in Resistance to Checkpoint Blockade Therapy 

Despite the presence of immunogenic antigens on cancer cells (i.e., neoantigens), the anti-tumor response can still be impaired if the antigen-presenting system does not work properly. This is the case of approximately 50% of melanomas and is caused primarily by the complete or partial loss of the HLA class I complex. Consequently, the decreased expression of HLA proteins and/or genes regulating antigens presentation correlates with a worse prognosis [31]. Complete or partial loss of HLA class I function is frequently caused by mutations in the β-microglobullin 2 gene (*B2M*), including ”hotspot“ mutations in exon 1 [31]. B2M is a crucial factor required for the assembly of all HLA class I complexes and also participates in folding and cell surface transport of antigens. Mutations in the *B2M* gene may appear at the early stage of melanoma development [32]. It was shown that the proper expression of B2M positively correlates with the overall survival of melanoma patients during immunotherapy [33]. Decreased expression of this gene in melanoma cells constitutes (together with other genes) a resistance gene profile that is associated with T cell exclusion and which predicts response to immune checkpoint inhibitors [34]. Also, the occurrence of alterations in this gene is approximately three times higher in non-responders than in responders to immune checkpoint blockade (CPB) [35]. Lack of functional B2M may also be responsible for acquired resistance to ICIs. Sucker at al. described clinical cases of melanoma patients treated and not treated with immunotherapy, whose progressed lesions evolved into complete loss of the *B2M* gene, which was accompanied by decreased infiltration of T cells due to the loss of HLA class I and II proteins [36]. A similar mechanism of resistance to immunotherapy was reported by Zhao et al. [37], Donia et al. [38], and in a colon cancer patient treated with adoptive T cell transfer [39]. Loss of MHC class I proteins in advanced melanoma was observed long before modern immunotherapy [40]. The aforementioned results concerning mainly B2M suggest that the HLA class I proteins are involved in resistance to both anti-CTLA-4 and anti-PD-1 therapy. However, some data shows that this group of HLA proteins may be predominantly responsible for the sensitivity of melanoma cells to anti-CTLA-4 treatment, while the expression of HLA class II molecules predicts response to anti-PD-1 therapy [41]. Further studies are required to confirm this interesting observation. Apart from *B2M*, other genes involved in antigen presentation with potential influence on response to therapy are *TAP1* and *TAP2* (coding for transporters associated with antigen processing) [33]. Loss of the TAP1 expression detected before treatment was significantly associated with the reduced overall survival of patients treated with ipilimumab and the reduced expression of TAP2 in melanoma cells promoted evasion of T-cell-mediated killing [33]. On the other hand, Chew et al. [42] identified DUX4 as a suppressor of HLA class I proteins. This is an embryonic transcription factor frequently re-expressed in cancer cells. It was shown that DUX4 expression in various cancer cells correlates with reduced expression of HLA class I genes including *B2M* and promotes resistance to immune checkpoints inhibitors in melanoma patients [42]. Interestingly, a statistically significant association was observed only for anti-CTLA-4 and not for anti-PD-1 therapy. This may stem from the small size of the cohort (as suggested by the Authors) or may be a true observation staying in line with the above-mentioned results of Rodig et al. [41]. 

Antigen presentation may also be impaired because of the dysfunctionality of the antigen preparation system, e.g., immunoproteasome, which degrades proteins before antigen presentation. Analysis of TCGA (The Cancer Genome Atlas) data revealed that the expression of two genes (*PSMB8*, *PSMB9*) coding for immunoproteasome proteins is a stronger immunotherapy predictive marker than TMB and both factors together had the best predictive capacity [43]. 

The results mentioned above show the importance of the antigen presentation system in an effective anti-melanoma immune response. Albeit necessary, the presentation of neoantigens would be useless if there were no immune cells in the tumor. 

## 5. “Cold” versus “Hot” Tumors and Response to Immunotherapy 

The response to immunotherapy depends also on the immune cells existing in tumor microenvironments such as lymphocytes, macrophages, dendritic cells, and myeloid derived suppressor cells (MDSCs) [16]. Tumors can be divided into three types depending on their immune topographies: (i) ”hot“ (“inflamed”), with a high infiltration of immune cells, (ii) ”cold” (“non-inflamed”), with a low amount of immune cells and (iii) “excluded”, with a high density of immune cells outside of the tumor and a low density inside [44]. Melanoma has been historically considered as an immunogenic malignancy and the presence of tumor infiltrating lymphocytes (TILs), defined as a polymorphic group composed mainly by effector T lymphocytes (Teffs), regulatory T lymphocytes (Tregs), natural killer (NK) cells, dendritic cells, and macrophages is a well described favorable prognostic factor [45,46]. However, the usefulness of the presence of immune cells in the tumor in predicting the clinical benefits of immunotherapy has not been fully elucidated so far. Some reports suggest that the presence of cytotoxic CD8+ T cells in the invasive margin of the tumor positively correlates with the response to pembrolizumab [47]. Deficient T cells trafficking to the tumor and impaired infiltration was also shown to be a significant obstacle to efficient immunotherapy [48]. However, the available data on the association of this parameter with the response to immune checkpoints inhibitors are not conclusive. Meta-analysis conducted by Jessurun et al. showed significant variations among various immune cells and therapeutic schemas [49]. It seems that the presence of active T cells may to some extent improve anti-PD1 response, while the presence of macrophages correlates with anti-CTLA-4 response [49]. However, further studies are needed to validate these observations. 

The situation is complicated by the presence of immunosuppressive cells in the tumor. Negative effect on anti-tumor activity of immune system is exerted by Tregs, myeloid derived suppressor cells (MDSCs), and tumor-associated macrophages (TAM, M2). Tregs are immunosuppressive cells that inhibit activity of Teffs by direct contact or indirectly by excreting specific cytokines (Il-10, Il-35, TGF-β) [50]. Eradication or reduction of Tregs in tumor microenvironment results in improved response to immunotherapy in some cancers [51,52]. Their role in melanoma immunotherapy is well established [53]. A recent study showed that inhibition of sphingosine kinase-1 (SK1) resulted in decreased Tregs trafficking to the tumor and consequently enhanced response to ICI in the murine model of various cancers including melanoma [54]. Depletion of Tregs was also associated with an improved efficacy of anti-CTLA-4 treatment combined with TLR1/2 (Toll-like receptor 1/2) ligands in the murine model of melanoma [55]. Other molecules that regulate Tregs activity are currently being investigated in preclinical studies [56] and tested in clinical trials in combination with checkpoint blocking agents [57,58]. 

Strong immune response inhibition is also mediated by highly immunosuppressive myeloid-derived suppressor cells. The presence of MDSC in the tumor correlates with worse response to various types of immunotherapy including immune checkpoints inhibitors [16,59]. A study on animal model revealed that the inhibition of tyrosine kinases highly expressed in these cells attenuates their immunosuppressive properties and increases the effectiveness of immunotherapy [60]. Of special interest are monocytes deprived of HL-DR antigens, which are responsible for such pathologic states as sepsis or tumor immunosuppression. An increased amount of these cells (CD14+HLA-DR^lo/neg^) in a patient’s blood is associated with impaired response to immunotherapy [61]. The MDSCs are probably recruited and expanded in the tumor by cytokines such as CXCL8/IL-8 and IL-6, respectively [62]. An increased level of one or both of them negatively correlates with melanoma patients’ prognosis [62]. 

## 6. Gene Signatures for Predicting Response to Immune Checkpoint Inhibitors 

The role of immune cells existing in the tumor microenvironment in immunotherapy responses was further proved at the gene expression level. The evaluation of expression profile of selected genes specifically expressed in T lymphocytes (e.g., granzyme B, perforin, chemokines: *CCL2*, *CCL3*, *CCL4*), macrophages (*CD68*, *CD14*), or dendritic cells (*BATF3*), may be a promising alternative for proper classification of ”hot” vs. ”cold” tumors [63]. In colon cancer, expression profile of immune-related genes (e.g., *IFNG*, *IL15*, *GNLY*, *CCL3*, *CXCL16*) together with infiltration of the tumor with cytotoxic and memory lymphocytes (“Immunoscore“) are prognostic factors independently of the presence of microsatellite instability (MSC) [64]. In melanoma, the expression profile of immune-related genes seems to correlate with the response to various immunotherapy schemas. The analysis of 45 patients revealed that those who responded to anti-CTLA-4 therapy had increased expression of most of the genes of 23-gene “immune” signature in tumors [65]. Also, the signature of 10 interferon-related genes elected by Ribas et al. showed predictive capacity in anti-PD-1 therapy. Increased expression of these genes was more frequently observed in patients who derived benefit from the treatment [66]. The expression level of some immune-related genes seems to be predictive also for the efficiency of adjuvant immunotherapy [67]. 

Furthermore, immune-related genes constituted gene signatures of potential predictive significance in immunotherapy identified by Poźniak et al. [68], and Hugo et al. [28]. In the first signature, expression of genes of various interferons pathways, antigen processing and presentation as well as TNF and chemokines pathways were significantly higher in the “high immune” subgroup of tumors in comparison to “low immune” tumors. More importantly, the immune status of the tumor correlated with overall survival. The IPRES (Innate anti-PD-1 Resistance) signature identified by Hugo et al. contained immunosuppressive genes (*IL10*, *VEGFA*, *VEGFC*) as well as monocyte and macrophage chemotactic genes (*CCL2*, *CCL7*, *CCL8*, and *CCL13*) [28]. This tumor signature characterized patients unresponsive to immunotherapy and had the capacity to predict overall survival. Immune-related genes, including interferon pathways, cytokines and chemokines pathways are also part of a predictive signature identified by Gide et al. [69]. Increased expression of these genes in tumors characterized patients who responded to anti-PD-1 or anti-PD-1 + anti-CTLA-4 therapies, while tumors of unresponsive patients demonstrated increased expression of the WNT pathway and, interestingly, hypoxia marker, carbon anhydrase 9 (*CA9*). A more complex approach to the problem was presented by Jiang et al. [70]. They developed a test based on gene expression signature (TIDE), which integrated both the expression signatures of T cell dysfunction and T cell exclusion. The test predicted response of melanoma patients to anti-PD-1 therapy. Genes pointed as the most interesting by the Authors are *PD-1*, *TGFB1*, *SOX10*, and *SERPINB9*. 

The aforementioned signatures are predominantly indicative of immune system activity in the tumors, however, they may also identify the activation of immunosuppressive pathways in melanoma cells, which may contribute to impaired lymphocyte trafficking and tumor infiltration. 

## 7. Melanoma Cell Signaling Pathways Associated with “Cold” Tumors 

In addition to the lack of and/or inability to present neoantigens on tumor cells, the reason for impaired anti-tumor response may also be the lack of tumor-infiltrating immune cells. The state of ”cold” tumors can be driven by cancer cells with activated signaling pathways that prevent the infiltration of immune cells into the tumor and modulate their topography. The signaling pathways of interferon, WNT/β-catenin, PI3K/AKT, and MAPK are best characterized in this respect. There are also other signaling pathways regulating immune response whose role in this complex process needs to be elucidated, e.g., MYC and NFκB pathways. The analysis performed by Poźniak et al. revealed that genetic changes in MYC and NFκB pathway genes differ between “hot” and “cold” tumors [68]. The “cold” tumors were characterized by more frequent deletions in NFκB pathway genes and amplification of MYC oncogene. The presence of these genetic alterations was associated with shorter overall survival. 

### 7.1. Interferon Pathway 

Interferon pathway is a key pathway modulating the effectiveness of immunotherapy [16]. Interferons are cytokines produced in response to pathogenic agents, mainly viruses. There are three types of interferons: I, II, and III. The anti-tumor response is predominantly regulated by type I and II. Type I interferons (e.g., interferon alpha) stimulate activity of dendritic cells and priming of lymphocytes T, while type II interferons (interferon gamma) affect both immune cells and tumor cells and facilitate elimination of the latter ones [71]. The key player is interferon γ secreted mainly by T lymphocytes. IFNγ activates both dendritic cells and macrophages to enhance antigen presentation [72]. INFγ signaling on tumor cells can also facilitate immune recognition and apoptosis of tumor cells [73,74]. That is why IFNγ signaling is so important in response to immunotherapy and its deregulation may result in a limited benefit from immune checkpoint inhibition. 

The impaired interferon signaling in tumor cells may result from mutations in genes regulating this molecular pathway. Such mutations are more frequently detected in patients who do not respond to immunotherapy [73]. Mutations are detected in both the main genes of the interferon pathway (loss of *IFNGR1*, *IRF1*, *JAK2*, *IFNGR2*) and genes of its inhibitors (amplification of *SOCS1*, *PIAS4*) [73]. Analysis of TCGA data revealed that 30% of melanoma samples harbor mutations in the INFγ signaling pathway and the presence of the mutations correlates with shorter overall survival. The importance of this pathway was further confirmed in in vitro and animal studies [73]. Genes which seem to play a significant role in resistance to immunotherapy are *JAK1* and *JAK2*. They code for tyrosine kinases, which activate interferon α/β pathway (JAK1) and interferon γ pathway (JAK1 and JAK2). Approximately 3–4% of the unselected melanoma patient population harbor mutations in these genes [75]. *JAK1/2* mutations are also identified in the tumors of patients who relapsed after immunotherapy [76]. Functional studies confirmed that the identified mutations (*JAK1^Q503^*, *JAK2^F547^*) affect the response of the cells to interferon. Cells were less sensitive to INFγ when mutated *JAK2* was present or to INFγ, α, and β in case of *JAK1* mutation [76]. Deletions of *JAK2* and other interferon γ pathway genes are also more frequently detected in tumors showing low immunological activity [67]. Also, loss-of-function mutations in the *APLNR* gene (leading to aberrant JAK1 action and INFγ responses) have been found in large screening studies as associated with the resistance to immunotherapy in melanoma patients [33]. Furthermore, an important role of INFγ in response to immunotherapy was supported by gene expression analyses in melanoma tumors before treatment with anti-CTLA-4 antibodies [65]. Expression of genes induced by interferon γ (e.g., *CXCL4*, *CXCL5*, *CXCL9*, *CXCL10*, *CXCL11*, and *IDO1*) was significantly higher in responders than non-responders and further increased during therapy. Similarly, responders to anti-PD-1 therapy had an increased expression of IFN-related genes (e.g., *IRF1*, *TNF*, *IFNG*, and *STAT1*) [69], while suppression of this pathway was associated with immune evasion and poor response to immunotherapy in melanoma patients [34]. 

On the other hand, IFNγ may also exert the opposite effect by modulating resistance of tumor cells to immune response. It stimulates the expression of PD-L1 on tumor and immune cells and induces adaptive mechanisms independent on PD-L1, e.g., epigenetic changes in tumor cells, which increase T cell inhibitory receptors (TCIR) expression [71,77]. IFNγ can also promote apoptosis of tumor-reactive CD8+ T cells [78]. 

### 7.2. WNT/β-catenin Pathway

WNT/β-catenin pathway is an important signaling pathway regulating immune resistance. It is a conservative pathway controlling embryonic development and cell homeostasis through the regulation of cell proliferation and polarity. Activation of this pathway in tumor cells is associated with lower tumor lymphocyte infiltration [79]. Gene expression profile analyses confirmed increased activation of β-catenin pathway in tumors with low or no immune activity (“cold” tumors) in comparison to those with immune cells infiltration (“hot” tumors). Forty-eight percent of “T-cell non-inflamed” tumors showed elevated expression of at least 5 out of 6 genes under the control of this pathway (*VEGFA*, *TCF12*, *MYC*, *TCF1*, *APC2*, *EFNB3*) in comparison to 3.8% of “T cell-inflamed” tumors. “Cold” tumors also were more likely to harbor mutations leading to WNT/β-catenin pathway activation (8% of samples with a gain of function mutation in catenin beta-1, *CTNNB1*, and 11% with loss of function mutations in *APC*, *AXIN1*, and *TCF1* genes coding for inhibitors of the pathway). Furthermore, functional studies confirmed the role of the WNT/β-catenin pathway in the attenuation of immune anti-tumor response and revealed the mechanism. It was proved that activation of this pathway impairs recruitment and activation of dendritic cells (CD103+), most likely because of the decreased expression of CCL4 chemokine (repressed by ATF3), and consequently impaired priming of T cells (the ability of T cells to recognize tumor antigens) [79]. In line with the aforementioned results are observations that the level of β-catenin (CTNNB1) is increased in “cold” tumors (with low immunologic activity) in comparison to “hot” tumors [68] and expression of *WNT5A* is higher in patients responding to anti-PD-1 therapy in comparison to non-responders [28]. Activation of the β-catenin pathway may also cause secondary resistance to immunotherapy. Trujillo et al. presented a case report of a patient who developed resistance to a multi-peptide vaccine combined with interleukin-12 (which is known to stimulate the production of IFNγ) due to the absence of CD8+ T cells infiltration which was linked to the activation of β-catenin pathway [80]. 

### 7.3. PI3K/AKT/mTOR Pathway 

The PI3K/AKT/mTOR pathway regulates many important cellular processes including metabolism, cell cycle, and cell survival. It is also involved in immunosuppression. This pathway has a natural inhibitor, PTEN, whose function is to limit activity of this pro-survival pathway. A complex study by Peng et al. demonstrated that loss of PTEN was connected with lower sensitivity of melanoma cells to cell death induced by T cells and reduced migration of T cells to tumors [81]. Moreover, analysis of 39 melanoma samples revealed that patients with loss of PTEN were more likely to progress on anti-PD-1 therapy than patients with wild type PTEN. PTEN status in the tumor also correlated with CD8+ T cell infiltration. This observation was additionally confirmed by analysis of melanoma TCGA dataset. The putative molecular mechanism of immune resistance determined by PTEN loss involves induction of VEGF, which has immunosuppressive properties [81]. The role of PTEN in response to immunotherapy was also confirmed by other analysis of clinical samples. Recurrent copy number loss of many genes, including *PTEN*, was observed with a higher frequency in patients not responding to anti-CTLA-4 and anti-PD-1 therapies, especially in the double non-responders group [82]. The acquired loss of *PTEN* may also contribute to secondary resistance to immunotherapy caused by ineffective T cells infiltration to tumor sites as it was described in a patient case report by Trujillo et al. [80]. 

### 7.4. MAPK Signaling Pathway 

MAPK (i.e., RAS-RAF-MEK-ERK) pathway plays an important role in promoting cell growth and proliferation and is also involved in the host immune response. The pathway is frequently activated in melanoma due to the BRAF^V600^ mutation. Constitutive activation of this pathway results not only in unrestrained melanoma proliferation but also in immune evasion [83]. Activated MAPK pathway exerts an immunosuppressive effect by modulating both immunocyte exclusion and dysfunction, as well as recruitment and differentiation of immunosuppressive cells [84]. The first effect is probably caused by the downregulation of tumor antigens and decreased recognition by immune cells, while the other by secretion of cytokines which stimulate trafficking and infiltration of immunosuppressive cells [83]. Inhibition of this pathway results in the decreased production of immunosuppressive cytokines such as IL-6, IL-10, and VEGF [85]. IL-1α is another interleukin produced by BRAF mutated melanoma cells. After release, it stimulates the expression of immunosuppressive ligands (PD-L1, PD-L2) on tumor-associated fibroblasts, which causes suppression of melanoma-specific CD8+ T cells [86]. It was also observed that the presence of the mutated form of this oncogene is associated with lower TIL (tumor infiltrating lymphocytes), while inhibition of BRAF or MEK results in an increased number of melanocyte antigens on melanoma cells (e.g., MART-1, gp-100) and their enhanced recognition by lymphocytes [87]. Oncogenic BRAF^V600E^ was also shown to stimulate priming and recruitment of immunosuppressive Tregs, which restrains CD8 T cell-mediated immune surveillance [88]. It is also responsible for recruitment of MDSC as restoration of these cells is observed in melanomas resistant to BRAF/MEK inhibitors [89]. The immunosuppressive properties of MAPK pathway is confirmed by observations that its inhibition results in the stimulation of immune response. This ”accidental“ effect may contribute to complete response during treatment with BRAF/MEK inhibitors [90]. On the other hand, some forms of mutated BRAF can be recognized by immune cells as neoantigens, which leads to prolonged response to immunotherapy (adoptive lymphocyte transfer) [91]. However, this phenomenon is extremely rare; most likely because of elimination of such cells by immune cells before they developed into a malignant tumor. 

## 8. Epigenetic Factors Involved in Resistance to Immunotherapy 

Epigenetic mechanisms are important regulators of many crucial processes including immune evasion and therapy resistance [92]. Although they have not been extensively explored as predictive markers, there are some reports on the role of DNA methylation and chromatin remodeling in the acquisition of resistance to immunotherapy [93]. It was shown that the global chromatin methylation level is associated with immune cell infiltration of various tumors, including melanoma, and this association is independent of mutation burden and aneuploidy. Methylation loss correlated also with the gene signature characteristic for immune evasion [94]. The correlation of methylation status with clinical parameters was observed as well. The low methylation level of the *CTLA-4* promoter was associated with a weak response to immune checkpoints inhibitors (both anti-CTLA-4 and anti-PD-1) and overall survival of melanoma patients [95].

In the context of resistance to immunotherapy, the histone deacetylases (HDACs) and EZH2 histone methyltransferase seem to be particularly important. They participate in the gene expression regulation by chromatin remodeling. Histone deacetylases remove acetyl groups on the histone tails leading to DNA condensation and preventing transcription. Loss of histone acetylation on regulatory regions of tumor suppressor genes (e.g., *CDKN2A*) leads to melanoma development [96]. Most likely, this mechanism may also regulate the expression of immune genes e.g., *PD-L1*. It was observed that inhibition of class I HDACs led to increased expression of PD-1 ligands in melanoma cells and improved efficacy of anti-PD-1 therapy in the animal model [97]. Similarly, EZH2 was proved to induce adaptive resistance to immunotherapy and its inhibition augmented anti-melanoma immunotherapy in the animal model [98]. Other reports also showed the therapeutic benefit of combined immunotherapy with HDACs and EZH2 inhibitors [93]. Therefore, HDACs and EZH2 can be considered as therapeutic targets rather than predictive markers. More data on EZH2 and DNA methylation targeting in the context of immunotherapy is presented by Emran et al. [99]. Furthermore, components of the PBAF chromatin regulatory complex (*PBRM1, ARID2*, and *BRD7*) were identified in large screening as responsible for melanoma cell resistance to killing by cytotoxic T cells. This complex reduces chromatin accessibility for IFNγ-inducible genes within tumor cells and inhibition of its components synergizes with checkpoint blockade therapy [100]. More information about epigenetic regulators of response to immunotherapy in various cancers and potential biomarkers of this type is presented in the review papers by Strub et al. [93] and Xiao et al. [101]. 

The described molecular factors are the main determinants of anti-tumor immune response and effectiveness of immunotherapy, but certainly, this manuscript did not exhaust the subject. The immune response is a complex process regulated not only by cancer cells but also by other factors such as hypoxia, polymorphisms, and microbiome. The detailed description of the current knowledge on these issues is beyond the scope of this review and below we only signal the importance of these biological factors in immunotherapy and refer the readers back to the most up-to-day review papers on these subjects. 

## 9. Hypoxia and Immunosuppression 

Hypoxia (decreased oxygen level) is an inherent factor of the tumor microenvironment, which inhibits anti-tumor response. Low oxygen tension causes immunosuppression through induction of immunosuppressive cells (Treg, MDSC, TAM) recruitment and influencing the activity of immune cells within the tumor [102]. Hypoxia impairs antigen presentation by dendritic cells and their ability to activate T cells. Moreover, low oxygen tension increases the expression of VEGF, IL-10, and metalloproteases. They inhibit dendritic cell maturation and weaken susceptibility of tumor cells to NK or T cells-induced lysis. Hypoxia also induces expression of PD-L1 on MDSCs [102]. More about the immunosuppressive properties of a hypoxic microenvironment is written in reviews by Noman et al. [102], and Daniel et al. [103]. 

## 10. Host Factors Influencing Response to Immunotherapy in Melanoma 

Host factors are the traits of a person that affect various physiological processes including susceptibility to disease or response to treatment. They encompass such determinants as gene polymorphisms (inherited genetic background), microbiome, sex, and age. Some reports suggest that age [104] and sex [105] may influence the response to immunotherapy. However, more data is required on this subject. Here, we will focus on two host factors: SNPs and microbiome. 

### 10.1. Single Nucleotide Polymorphisms 

Host factors such as single nucleotide polymorphisms (SNPs) can affect the response to immunotherapy. Analysis of 25 polymorphisms associated with autoimmune diseases of 436 patients revealed the strong association of at least one SNP with the response to anti-PD-1 therapy [106]. Other studies suggest that some polymorphisms may modulate the response to immunotherapy by regulation of immune genes expression (e.g., *ERAP2*, *ICOSLG*). Also, polymorphisms of HLA genes may affect the effectiveness of immune checkpoints inhibition through impaired antigens presentation and cause poor response despite high TMB [106]. Similarly, the presence of some polymorphic versions of *PD-1*, *PD-L1,* and *CTLA-4* genes is associated with the response to immune checkpoints inhibitors [107]. Immunopharmacogenomics is a promising but relatively young branch of science [108]. Further Genome-Wide Association Studies (GWAS) are required to elucidate the exact role of SNPs in response to immunotherapy. 

### 10.2. Microbiome 

The microbiome is the entire collection of microorganisms inhabiting our organism, including gut microflora, which comprises approximately 3 x 10^13^ microorganisms [109]. It is an especially important factor influencing the functionality of our whole body including the immune system and its anti-tumor activity [110]. Animal studies, as well as clinical observations, suggest that both the quantity and quality of the gut microflora affect cancer development and anti-tumor immune response [111]. It was shown that antibiotic therapy could change the gene expression profile in animal tumors. It increased the expression of “pro-tumor” genes and decreased those regulating antigen presentation, inflammation, or phagocytosis. It was also observed that tumors growing in animals deprived of their microflora had small neutrophil and monocytes infiltration and showed poor response to immunotherapy. The putative reason was the low number of TNF-producing immune cells in the tumor [112]. The influence of gut microbiota on anti-tumor response was also observed by Tanoue et al. [113]. The introduction of 11 commensal species of human gastrointestinal bacteria (including 7 species of *Bacteroidales*) into mice resulted in a strengthening of immune response to bacterial infections and anti-tumor immune response (spontaneous and induced by immunotherapy). The effect was mediated by the induction of IFNγ-producing CD8+ T cells in the gut and increased infiltration of the tumor by these cells. The positive influence of *Bacteroidales* species on the efficiency of immunotherapy in mice was also observed by Vetizou et al. [114]. The efficiency of immunotherapy in mice deprived of gut microbiota was restored after introducing *B. fragilis* bacteria or immunization with their lipopolysaccharides or after adoptive transfer of *B. fragilis*-specific T cells. 

Human studies support the data obtained using animal models. It was shown that a more diverse microbiome enriched with *Faecalibacterium* and *Ruminococcaceae* bacteria bolster the immune system and anti-tumor response through enhancing antigen presentation and activity of lymphocytes in the tumor [115]. It was also observed that gut microbiota differs between responders and non-responders to immunotherapy (with *Bifidobacteriaceae* dominated in responders). The immunogenic effect of the microbiota derived from responders was further confirmed in the murine model [116]. Our microbiota may also negatively influence the anti-tumor immune response. Ma et al. have shown that metabolic activity of *Clostridium* bacteria impairs anti-tumor response, which was manifested by an increased number of metastatic lesions in murine liver [117]. 

The influence of the gut microbiome on the response to immunotherapy is relatively well established. The putative mechanisms of this phenomenon are the stimulation of T cells activity, induction of expression of receptors mediating anti-tumor and inflammatory response, and systemic activation of the immune system by specific microbial metabolites. However, further research is needed to identify and characterize the specific microbiota profile, which unquestionably modulates the anti-tumor activity of the immune system and correlates with response to immunotherapy in melanoma patients. This is of great importance not only to be able to predict a patient’s response, but also to enhance it by microbiome modifications. 

## 11. Summary 

Immunotherapy with immune checkpoint inhibitors has revolutionized the treatment of advanced melanoma patients and has given a chance for prolonged survival to many of them. However, this therapy has some serious limitations. Foremost, a significant number of patients do not respond to the treatment. Moreover, patients who qualify for this treatment need to have a good or very good baseline performance status since immunotherapy needs time for clinical response. Furthermore, immunotherapy may cause severe adverse effects, and finally, the cost is high. Therefore, it is so important to be able to predict the efficacy of this treatment before its initiation and qualify only patients with the highest probability of response. This can be achieved with the usage of predictive markers, which are currently intensively sought-after among biological factors that influence the anti-tumor immune response. These factors can be divided into three groups: associated with melanoma cells, linked to the tumor microenvironment, and associated with the whole organism (Figure 1 and Figure 2). Some of them have already been established as potential predictive biomarkers, while others need further investigation (Table 1). The huge amount of basic research as well as clinical studies focused on searching for predictive markers for immunotherapy gives hope that the multi-factorial test will soon be available for broad clinical testing. 

## Figures and Tables

**Figure 1 ijms-21-04071-f001:**
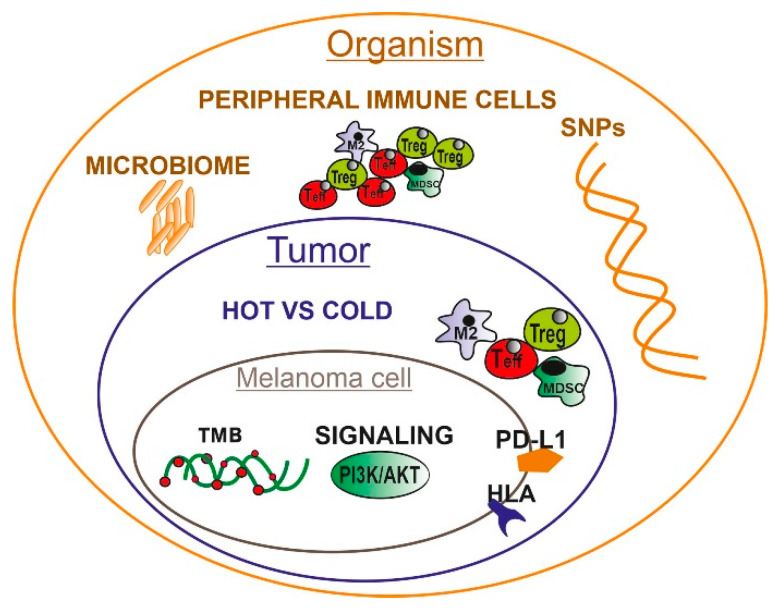
A nested hierarchy of biological factors influencing the anti-tumor immune response at the cell, tumor, and body levels. On a melanoma cell level, the immune response depends on such factors as tumor mutational burden (TMB), expression of the ligand of PD-1 molecule (PD-L1), and HLA proteins responsible for neoantigens presentation. On the tumor level, the anti-melanoma response is regulated by the presence of immune cells (“hot” tumors vs. “cold” tumors) and on the level of the organism by the microbiome, inherited profile of single nucleotide polymorphisms (SNPs) and status of the peripheral immune system.

**Figure 2 ijms-21-04071-f002:**
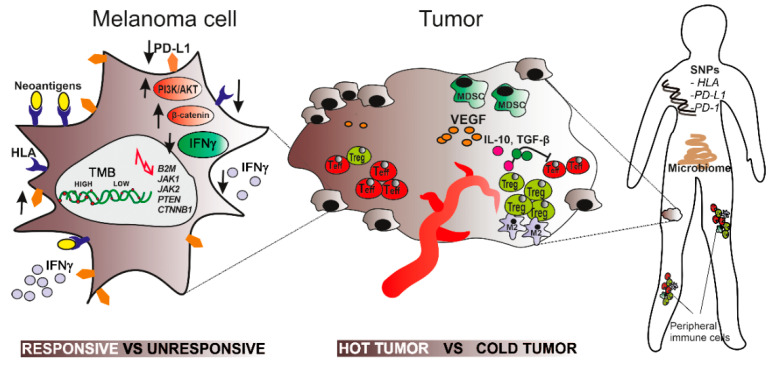
The graphical summary of biological factors influencing anti-tumor immune response constituting three categories depending on anatomical level: tumor cell, tumor microenvironment, and the whole body. Melanoma cells influence anti-tumor response passively (through the number of mutations and neoantigens) and actively (by regulating some key signaling pathways). The unresponsive cell (right part of the schematic cell) is characterized by a low tumor mutational burden (TMB), low expression of neoantigens, HLA molecules, and PD-L1 ligand (indicated with downward arrows) as well as activated β-catenin and PI3K/AKT pathways and a suppressed IFNγ pathway (indicated with upward and downward arrows, respectively). Deregulation of these pathways results in the secretion of specific cytokines (e.g., VEGF, IL-10, TGF-β). They induce immunosuppressive topography of the immune cells within the tumor, which is characterized by a low density of the cytotoxic immune cells and the predominance of the immunosuppressive cells (Tregs, MDSC) (the right part of the schematic tumor). On the whole body level, the response of melanoma to immunotherapy is modulated by the specific gut microflora, constellation of peripheral immune cells, and inherent genetic variants of some important proteins e.g., HLA. The response of the tumor to the immune checkpoints inhibitors as well as the success of the immunotherapy depends on all those biological factors.

**Table 1 ijms-21-04071-t001:** Potential predictive biomarkers in melanoma immunotherapy (ordered from the most promising and most extensively studied).

Biomarker (DNA, mRNA, Protein)	Mechanism of Sensitivity/Resistance	Predictive Capacity: Pros and Cons, Perspectives	References
PD-L1 (protein)	Ligand for immune checkpoint molecule (PD-1). Their interactions inhibit activity of cytotoxic T cells	Association with the response to immunotherapy, but low specificity and sensitivity as a predictive marker; lack of standardized assay; unsatisfactory negative and positive predictive values	[13,14,15]
TMB—Tumor mutational burden (DNA)	New tumor-associated antigen recognized by immune cells	High tumor mutational burden (TMB) increases the probability of good response to immunotherapy, but does not guarantee it; the assay should be cancer type-specific; the number of sequenced genes should be established; the assay is prone to technical parameters e.g., variant calling methodology, cut-off criteria	[23,24,25,26,27,28,29]
*B2M* (DNA, mRNA)	Nonfunctional antigen presentation due to impaired synthesis and transport of MHC class I proteins	β-microglobullin 2 gene (B2M) expression positively correlates with survival during immunotherapy; loss—may lead to secondary resistance; it should be a part of genetic predictive panel	[33,34,35,36,37,38]
*PTEN* (DNA)	Resistance to T cell- induced apoptosis; decreased T cell infiltration	Higher frequency of PTEN loss in non-responding patients; possible role in secondary resistance; it should be a part of genetic predictive panel	[80,81,82,118,119,120]
*JAK1, JAK2* (DNA)	Insensitivity to INFα, β, γ (*JAK1*) and INFγ (*JAK2*)	Mutations are identified in relapsed samples; larger genetic analyses are needed to evaluate the predictive capacity; it should be a part of genetic predictive panel	[75,76]
Gene expression profiling of the tumors (mRNA)	Differential expression of immune genes	Distinguishing between ”hot“ and ”cold“ tumors; a potential predictive capacity that should be further validated	[28,63,64,65,66,67,68,69,70]
*CTNNB1* (DNA, mRNA)	Activation of the β-catenin pathway prevents lymphocyte infiltration	Activation of β-catenin pathway and expression of CTNNB1 is higher in tumors with low immune cell infiltration; more data required; genetic analysis of *CTNNB1* should be a part of genetic predictive panel	[68,79,80]
Interferon pathway genes (e.g., *IFNGR1*), (DNA)	Impaired interferon pathway	Higher frequency of loss or mutations in non-responding patients; more data required but they should be a part of genetic predictive panel	[73]
VEGF (mRNA, protein)	Immunosuppressive cytokine	A part of IPRES signature identified by Hugo et al. [28]; elevated expression in “cold” tumors; too low predictive specificity possible due to pleiotropic activity	[28,79]
CNA (Copy number alterations)	Loss of tumor suppressor genes including *PTEN*; decreased activity of immune signaling pathways	Loss of copy number of 6q, 10q, 11q23.3 in double non-responders; more data required	[82]
*TAP1* (mRNA)	Impaired lymphocyte activity	Patients with increased expression respond better to immunotherapy; too little data to evaluate the predictive capacity	[33]
*APLNR* (DNA)	Regulation of JAK-STAT signaling pathway; impaired response to INFγ	Mutations detected in tumors refractory to immunotherapy; more data required	[33]
*PBRM1, ARID2 BRD7* (mRNA)	Increased sensitivity of melanoma cells to INFγ and T cell-stimulated apoptosis	Correlation of expression with survival in patients with higher CD8 expression; more data required (*ARID2*)	[100]
Amplification of *MYC* and deletion of NFκB pathway genes	Immune evasion, suppression of lymphocytic infiltration	Alterations present more frequently in “cold” tumors of non-responders (shorter overall survival); more data required	[68]

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
