# Peer review of "Biological Factors behind Melanoma Response to Immune Checkpoint Inhibitors"

_ijms, 2020, doi:10.3390/ijms21114071_

Round 1

Reviewer 1 Report

The review on "Biological factors behind melanoma response to immune checkpoint inhibitors" is quite comprehensive and well laid out. It covers all the pertinent topics and provide in detail analysis of supporting and contradicting positions in the literature. Here are minor recommendations which could benefit the review

  1. Section about epigenetic factors. This is glaring as the authors have extensively covered the genetic factors including mutations and SNPs. There is emerging evidence on silencing of immune response genes by either DNA methylation or chromatin regulation by HDACs 
  2. The authors have aptly summarized a table for potential biomarkers to consider which is impressive based on the arguments presented in the review, but this also made the review limited to those genes. The reason is predominant discussion of the review was around these genes and others were minimized. For example, B2M was the entire topic in the antigen presentation section. TAP1 and TAP2 were barely mentioned. There other topics such as cross presentation of antigens which directly activated CD8 T-cells. I would encourage the authors to expand the scope of sections if possible.
  3. Any information on upcoming interests in other immune checkpoint inhibitor currently under consideration is missing. The review is Anti-PD1 centric and I understand the authors provided their justification in the section 2. However, it would be such a disservice to not mention other potential immune checkpoint modulators considering how well the review is presented. 

Author Response

Dear Reviewer,
We would like to thank the Reviewer for insightful comments on the manuscript and suggesting a minor revision. Below we provide point-by-point response (in red) to the raised queries.

1. Section about epigenetic factors. This is glaring as the authors have extensively covered the genetic factors including mutations and SNPs. There is emerging evidence on silencing of immune response genes by either DNA methylation or chromatin regulation by HDACs
Ad. 1. Apologies for this omission. The chapter “Epigenetic factors involved in resistance to immunotherapy” was added to the manuscript according to suggestion (chapter 8 in the revised manuscript).

2. The authors have aptly summarized a table for potential biomarkers to consider which is impressive based on the arguments presented in the review, but this also made the review limited to those genes. The reason is predominant discussion of the review was around these genes and others were minimized. For example, B2M was the entire topic in the antigen presentation section. TAP1 and TAP2 were barely mentioned. There other topics such as cross presentation of antigens which directly activated CD8 T-cells. I would encourage the authors to expand the scope of sections if possible
Ad. 2. The section about antigen presentation (chapter 4) was supplemented with more data on this issue.

3. Any information on upcoming interests in other immune checkpoint inhibitor currently under consideration is missing. The review is Anti-PD1 centric and I understand the authors provided their justification in the section 2. However, it would be such a disservice to not mention other potential immune checkpoint modulators considering how well the review is presented.
Ad. 3. The information about other agents of immunotherapy which are being tested was added to the “Immunotherapy of melanoma” section (line 44 of the revised manuscript, chapter 1).

Reviewer 2 Report

Biological factors behind melanoma response to immune checkpoint inhibitors

May 2020

This manuscript is an up-to-date review of recent literature describing the factors that contribute to immune checkpoint inhibitor efficacy.  Overall, this review is well written, clearly defines its scope, and nicely offers the authors’ perspective on various topics.  This review topic is somewhat saturated in recent years – however given the gravity of the subject, continued commentary/updating is warranted.  Overall, this review is well done -- please see below for specific comments:

Major concerns:

  1. In general, the figures and the table are not clear and it is difficult to understand how they add to the manuscript. Are the concentric circles in Figure 1 a representation of hierarchical nesting?  Figure 2 initially looks interesting, however it is not clear what the arrows represent and if highlighted components in each setting (melanoma, tumor, or patient) are positive or negative prognostic indicators.  Table 1 seems disorganized.  What is the ordering? Why have genes listed in the “Potential predictive biomarker in melanoma immunotherapy” table when there is “No Data” for “Predictive capacity…”?  What were the criteria for being listed? For instance VEGF ligands were discussed multiple times in the text as a potential indicator, but were not listed.
  2. The authors did not comment on Kugel et al. Clin Can Res 2018 (aging) as a prognostic determinant for therapy efficacy.
  3. Authors may want to mention the link between the inflammatory response of dead/dying melanoma due to targeted inhibitors to immunotherapy efficacy. Erkes et al. Can Discov. 2020, perhaps -- Amaria RN, Lancet Oncol. 2018.
  4. Authors may want to discuss pros/cons of dosing schedules of combined targeted inhibitors with immunotherapy as this pertains to “cold” and “hot” premise.

Minor concerns:

  1. The word “organism” in the abstract is maybe a little awkward.
  2. There are multiple instances of “…there are these categories: First,… second, …“ this phrasing multiple times in the manuscript was off-putting.
  3. The paper may benefit from improved demarcating/separating of the predictive TMB studies with the non-predictive/significant studies.

Author Response

Dear Reviewer,
We would like to thank for insightful comments on the manuscript and suggesting a minor revisions. Below we provide point-by-point response (in red) to the raised queries.
Major Concerns:
1. In general, the figures and the table are not clear and it is difficult to understand how they add to the manuscript. Are the concentric circles in Figure 1 a representation of hierarchical nesting? Figure 2 initially looks interesting, however it is not clear what the arrows represent and if highlighted components in each setting (melanoma, tumor, or patient) are positive or negative prognostic indicators. Table 1 seems disorganized. What is the ordering? Why have genes listed in the “Potential predictive biomarker in melanoma immunotherapy” table when there is “No Data” for “Predictive capacity…”? What were the criteria for being listed? For instance VEGF ligands were discussed multiple times in the text as a potential indicator, but were not listed.
Ad. 1. The aim of the figures is to sum up and visualize the groups of biological factors described in the paper on general (Figure 1) and more detailed (Figure 2) level. To make them clearer and more comprehensive the descriptions of the figures were corrected according to the remarks.
Thank you for the comment on the Table. The Table was reorganized and the presented biomarkers were ordered according to their potential for predicting the efficacy of immunotherapy.
2. The authors did not comment on Kugel et al. Clin Can Res 2018 (aging) as a prognostic determinant for therapy efficacy.
Ad. 2. The response to immunotherapy is a complex process regulated by multiple factors. In our review, we focused on the factors associated predominantly with melanoma cells and those with the highest potential for use in clinics as predictive biomarkers. Obviously, age is one of the factors which may influence response to immunotherapy, however it did not meet our selections criteria. Therefore, it was only mentioned in the article along with other host factors (chapter 10).
3. Authors may want to mention the link between the inflammatory response of dead/dying melanoma due to targeted inhibitors to immunotherapy efficacy. Erkes et al. Can Discov. 2020, perhaps -- Amaria RN, Lancet Oncol. 2018.
4. Authors may want to discuss pros/cons of dosing schedules of combined targeted inhibitors with immunotherapy as this pertains to “cold” and “hot” premise.
Ad. 3 and 4. Though interesting, the issues of interaction between targeted therapy and immunotherapy addressed by the Reviewer are beyond the scope of this review article.
Minor concerns:
1. The word “organism” in the abstract is maybe a little awkward.
Ad. 1. The word “organism” was changed to “patient” (line 18 of the revised manuscript)
2. There are multiple instances of “…there are these categories: First,… second, …“ this phrasing multiple times in the manuscript was off-putting.
Ad. 2. The English was corrected accordingly by a native speaker.
3. The paper may benefit from improved demarcating/separating of the predictive TMB studies with the non-predictive/significant studies.
Ad. 3. The aim of this review paper was to present the most up-today knowledge of biological factors influencing response of melanoma patients to immunotherapy with a special focus on melanoma-associated determinants and also those with the most promising predictive capacity. However, predictive capacity was not the main criteria for selection of the factors being described. That is why in the text the factors are not divided into “predictive” and “non-predictive” studies. Still, in the Table we decided to order the potential biomarkers according to their potential for predicting response to immunotherapy.